# Recent Advances in Technologies for Preserving Fresh-Cut Fruits and Vegetables

**DOI:** 10.3390/foods14162769

**Published:** 2025-08-09

**Authors:** Muhammad Faisal, Naeem Arshad, Hui Wang, Chengcheng Li, Jinju Ma, Xiaoxue Kong, Haibo Luo, Lijuan Yu

**Affiliations:** 1Agro-Products Processing Research Institute, Yunnan Academy of Agricultural Sciences, Kunming 650221, China; naeem.ali01122@gmail.com (M.F.); wanghui_jgs@163.com (H.W.); 2School of Food Science and Pharmaceutical Engineering, Nanjing Normal University, Nanjing 210023, China; naeemarshad306@gmail.com (N.A.); kongxiaoxue@163.com (X.K.); 3International Innovation Center for Forest Chemicals and Materials, Jiangsu Co-Innovation Center for Efficient Processing and Utilization of Forest Resources, Nanjing Forestry University, Nanjing 210037, China; lichengcheng@njfu.edu.cn; 4Institute of Highland Forest Science, Chinese Academy of Forestry, Kunming 650233, China; majinjuchem@163.com

**Keywords:** fresh-cut produce, quality deterioration, microbial contamination, nutrient loss, modern preservation technologies

## Abstract

Rapid economic growth and changing consumer patterns have made fresh-cut fruits and vegetables household staples because of their high nutritional value, their role in reducing the risk of illnesses and other health problems, and convenience. However, fresh-cut produce is susceptible to the rapid deterioration of sensory quality, nutrient loss, foodborne pathogens contamination, and spoilage caused by microbial growth, which can lead to consumer health risks. Thus, there is an urgent need to improve preservation methods, to increase the shelf life of fresh-cut produce. This review examines the primary mechanisms underlying quality deterioration in fresh-cut produce and critically evaluates emerging preservation technologies including physical, chemical, and biopreservation for their efficacy in reducing microbial growth while maintaining product quality. This paper also discusses key gaps and proposes future research directions to improve preservation methods, extend shelf life, and ensure the safety of fresh-cut produce.

## 1. Introduction

Fresh-cut fruits and vegetables (FCFVs) are popular because of the increased demand for healthy, convenient, ready-to-eat food products. Globally, health-conscious consumers prefer fresh-cut produce for its flavor, nutritional content, and freshness [1]. However, FCFVs undergo a series of processing steps, such as peeling, slicing, chopping, and cutting, which damage tissues and accelerate quality deterioration [2]. The fresh-cut process causes mechanical damage, exposing the cellular structure of produce, increasing respiration rates, and causing moisture loss, which leads to microbial contamination [3]. Enhanced decay causes tissue softening, enzymatic browning, nutrient loss, and flavor change. Foodborne viruses and spoilage bacteria reduce the shelf life of FCFVs and can be hazardous to human health, causing outbreaks of foodborne diseases [4]. Consequently, methods for extending the shelf life of FCFVs should be established.

Scientists and manufacturers have been trying to develop techniques to preserve the quality and prolong the shelf life of products. Methods for preserving FCFVs can be classified as chemical, physical, and biological/biopreservative [5]. Physical preservation methods prevent spoilage and microorganism growth by changing ambient parameters, such as temperature, humidity, and gas composition [6,7]. Chemical preservation methods may use either natural or synthetic preservatives that delay oxidation and may also inhibit microorganisms that cause nutrient degradation and off-flavors [8]. Nevertheless, consumer uncertainty about synthetic additives is increasing due to food safety and health concerns [8]. Therefore, natural preservatives that incorporate essential oils, organic acids, and other plant extracts are being produced. Biopreservation uses naturally occurring microorganisms and their antimicrobial products and has emerged as a promising alternative to synthetic preservatives. By leveraging the antibacterial properties of bacteriocins, bacteriophages, and bioprotective cultures, biopreservation techniques can inhibit the growth of spoilage organisms while also being perceived as a more natural and safe method of food preservation [9]. Despite the substantial progress in developing preservation methods, the limited sensory quality and shelf life of fresh-cut produce remain a challenge for the industry. Current research is focused on developing innovative preservation methods that satisfy the consumer demand for additive-free, minimally processed foods, while simultaneously ensuring safety and quality.

No one method effectively reduces the overall decline in quality. Various strategies are employed to enhance the sensory quality and extend the shelf life of FCFVs by slowing their deterioration rate. This paper is a comprehensive overview of current advances in preservation technologies for FCFVs and outlines the challenges and future directions.

## 2. Reasons for Quality Deterioration of Fresh-Cut Fruits and Vegetables

The physiology of fresh-cut food items includes the biochemical changes that occur in the tissues of plants that have been injured. Various wounding signals can arise as a result of abiotic stressors due to external mechanical damage, such as cutting and peeling. Salicylic acid, ethylene, abscisic acid, calcium ions (Ca^2+^), methyl jasmonate, and other wounding signals spread from wounded tissues into adjacent non-injured tissues, causing physical damage that must be reduced to preserve fresh-produce quality [10]. Cutting damages tissues mechanically, altering respiration and transpiration, increasing susceptibility to microbial infection, and accelerating senescence; these changes lead to deterioration that affects the color and taste of the product, limiting the shelf life even when stored under cold chain conditions [11]. Figure 1 summarizes the possible reasons for the quality deterioration of FCFVs.

### 2.1. Enhanced Ethylene Production

The phytohormone ethylene directly affects the shelf life and freshness of FCFVs. Physiologically, it controls ripening, senescence, and stress responses. It catalyzes enzymatic activity that accelerates color degradation, softening, and the resulting loss of texture. Its impact is more distinct post-harvest, since mechanical injury increases ethylene production [12]. Fruits can be categorized as climacteric fruits, which show an intense spurt in ethylene production at the ripening stage, and non-climacteric fruits, such as cucumbers and strawberries, which produce less ethylene but are sensitive to exogenous ethylene, increasing quality degradation [5].

Ethylene biosynthesis, like many other reactions, responds to wound- or damage-initiated stimuli via the enzyme-catalyzed conversion of S-adenosylmethionine into 1-aminocyclopropane-1-carboxylic acid, which is then oxidized into ethylene [12]. The biosynthetic pathway is rapidly activated, as cutting the fruit increases ethylene synthesis, causing it to brown and soften quickly. The efficient regulation of ethylene biosynthesis is needed to maintain the freshness of cut produce. Reducing ethylene activity through cold storage, ethylene scavengers, specific inhibitors (e.g., 1-methylcyclyopropene), and package-related ethylene-scavenger-based advances contributes to extended shelf life [13].

### 2.2. Enhanced Respiration Rate

Fresh-cut processing induces a cascade of physiological stress responses in plant tissues, among which the dramatic elevation of the respiration rate represents a critical postharvest challenge. This wound-induced metabolic surge occurs as injured cells accelerate their energy metabolism to fuel repair mechanisms, ultimately depleting carbohydrate reserves and accelerating senescence [14]. The respiration rate of fresh-cut yams increased from 60 mg of CO_2_ kg^−1^ h^−1^ before cutting to 210 mg of CO_2_ kg^−1^ h^−1^ (4 °C) after cutting [15]. The respiration rate of whole crown pear is about 6 mg of CO_2_ kg^−1^ h^−1^, which increases to 25 mg of CO_2_ kg^−1^ h^−1^ immediately after cutting [16]. The effect of different cutting methods on the respiration rate also differs. The finer the cuts, the larger the surface area of fruits and vegetables, and the higher the respiration rate. After cutting cabbage into 3, 1.5, 0.7, and 0.3 cm slices, the respiration rate increased from 23 mg of CO_2_ kg^−1^ h^−1^ before cutting to 51, 125, 133, and 194 mg of CO_2_ kg^−1^ h^−1^ after cutting, respectively [7].

The respiration rate is important in the degradation of the freshness of FCFVs, as it promotes a chain of biochemical and physiological processes that increase the spoilage rate [3]. A low respiration rate conserves the stored sugars and starches, which are crucial for maintaining cell functions and structural entities in tissues [17]. With the depletion of these energy reserves, tissues become more susceptible to oxidative stress and enzymatic breakdown. Reactive oxygen species (ROS), such as superoxide anions and hydrogen peroxide, are produced at enhanced respiration rates and can damage cellular structures like lipids, proteins, and deoxyribonucleic acid (DNA) [17]. Therefore, the tissue loses its crispness and becomes soft, an excellent indicator of deterioration.

### 2.3. Increased Transpiration

Similarly to whole vegetables and fruits, the water content of fresh-cut produce ranges from 80% to 95%. During cleaning and cutting, plant tissue transpiration is enhanced when protective structures such as the wax layer, cuticle, and transpiration pores break down [18]. A reduction in turgor caused by water loss of 4–6% of the mass of FCFVs causes tissue wilting and shrinkage. Higher concentrations of H^+^ and NH_4_^+^ ions in intracellular fluid destroy protoplasmic structures; hydrolases break macromolecules in cells into small molecules required for respiration; and higher ethylene and abscisic acid levels accelerate tissue senescence [19].

### 2.4. Increased Color Loss

#### 2.4.1. Cut Surface Browning

The most noticeable color reaction in fruits and vegetables is the brown and black pigments that form after the tissue has been exposed to air. The discoloration reaction is primarily catalyzed by polyphenol oxidase (PPO), peroxidase (POD), and phenylalanine ammonia-lyase (PAL), plus phenolase, diphenol oxidase, tyrosinase, and monophenol oxidase [11]. Since there are many cell walls in plant tissues, the release of phenolic compounds with which the enzymes interact upon wounding, particularly catechins and other polyphenols, leads to instant oxidation in the presence of oxygen [20]. PPO and POD catalyze these phenolic compounds into highly reactive quinones, which polymerize to form brown pigments called melanin. Although a plant resistance mechanism, this reaction results in undesirable discoloration on the cut surface of produce. Along with browning enzymes, oxygen and phenolic compounds participate in browning reactions (Figure 2A).

AOX, alternative oxidase; APX, ascorbate peroxidase; AsA, ascorbic acid; CAO, chlorophyllide a oxygenase; CAT, catalase; CHLG, chlorophyll synthase; Chl-POX, chlorophyll-degrading peroxidase; CLH, chlorophyllase; Cyt c, cytochrome c; DAHP, 3-deoxy-D-arabino-heptulosonate-7-phosphate; DHA, dehydroascorbic acid; DHAR, dehydroascorbic acid reductase; DOPA, dihydroxyphenylalanine; E4P, erythrose-4-phosphate; EMP, Embden–Meyerhof–Parnas pathway; GalDH, L-galactono-γ-lactone dehydrogenase; GluRS, glutamyl-tRNA synthetase; GluTR, glutamyl-tRNA reductase; GR, glutathione reductase; GSAM, glutamate-1-semialdehyde 2,1-aminomutase; GSH, reduced glutathione; GSSG, oxidized glutathione; H_2_O_2_, hydrogen peroxide; LOX, lipoxygenase; MDA, malondialdehyde; MDCase, Mg-dechelatase; MDHA, monodehydroascorbic acid; MDHA, monodehydroascorbic acid; MDHAR, monodehydroascorbate reductase; MDHAR, monodehydroascorbate reductase; NADPH, nicotinamide adenine dinucleotide phosphate; NOL, Chlorophyll(ide) b reductase; NTR, thioredoxin reductase; O_2_^·−^, superoxide anion; PAL, phenylalanine ammonia lyase; PEP, phosphoenolpyruvate; PLD, phospholipase D; POD, peroxidase; PPH, pheophytinase; PPO, polyphenol oxidase; PPP, pentose phosphate pathway; Q, ubiquinone; SOD, superoxide dismutase; TCA, tricarboxylic acid; TRX, thioredoxin; XOD, xanthine oxidase.

#### 2.4.2. Green Color Loss

The primary components that give fruits and vegetables their green hue are chlorophylls a and b, which deteriorate during preparation and storage, causing the green color to fade to yellow. Typically, fruits are picked before they are completely ripe, and their green color will decrease with storage [21]. This breakdown of chlorophyll occurs primarily via phenophytinase and chlorophyllase. Pheophorbide-a, the direct precursor of the colorless product, is created when chlorophyll is stripped of its phytyl and magnesium ions by the enzymes chlorophyllase and pheophytinase, respectively [22].

Under acidic conditions, chlorophyll is converted to pheophytin, which has a drab olive-green color. Slicing and peeling cause mechanical damage to fruits and vegetables, exposing plant cells to oxygen, which increases oxidative stress and speeds up color loss (Figure 2B). For instance, sliced cucumbers, asparagus, lettuce, and kiwifruit lose their green color when exposed to light and air because of enhanced oxidative reactions [23].

### 2.5. Increased Texture Loss

The loss in texture of FCFVs is brought about by the enzymatic degradation of cell wall components and membrane disruption. Aging, weight loss, senescence ceases, osmotic solute leakage, reduced turgor, wounding effects, and increased cell wall hydrolase activity are all linked to tissue softening. Cutting damages cell walls and releases enzymes like cellulase and polygalacturonase [24]. Polygalacturonase hydrolyzes α-1,4-glycosidic bonds in pectin, which weakens the middle lamella and causes tissue softening. Cellulase further weakens the walls by cleaving β-1,4-glycosidic bonds, thereby degrading cellulose [4]. Oxidative damage also enhances the loss of texture, and the ROS generated during cutting degrade lipids, proteins, and cell wall polymers, destabilizing membranes and leading to the leakage of cellular contents, changing osmotic pressure and turgor, resulting in softening [22]. Ethylene speeds senescence by stimulating pectinmethylesterase and polygalacturonase, further degrading pectin. Water loss from vacuoles reduces turgor pressure, causing further cell wall collapse and tissue softening [25].

In addition, tissue lignification, the rapid epimerization and polymerization of cell wall micro-fibrils due to excessive lignin accumulation, also affects the texture of FCFVs via cell wall thickening and reduces the appearance and flavor of FCFVs [14]. Tissue lignification is common in fresh-cut celery, cilantro, *Zizania latifolia*, carrots, asparagus, and bamboo shoots. Lignin is produced via the phenolic metabolism pathway, linked to the activity of enzymes of the phenylpropanoid metabolic pathway like PAL, 4-Coumarate-CoA ligase (4 CL), cinnamyl alcohol dehydrogenase (CAD), and POD.

### 2.6. Microbial Spoilage

Cutting, peeling, and bruising disrupt the natural barriers inside fresh-cut produce, damaging cells and releasing nutrients for microorganisms, such as amino acids and sugar. Physical injury leads to cytoplasmic leakage, increasing moisture, an essential condition for microbial colonization [11]. This increases the susceptibility of FCFV tissues to microbial contamination by bacteria (*Pseudomonas*, *Enterobacter*, and *Listeria monocytogenes*), yeasts, and molds [26].

Microbes multiply more rapidly in fresh-cut barely processed food than in intact produce [3,11]. For example, cut lettuce leaves often have higher microbial counts than intact heads and whole leaves [24,27]. The increased microbial populations also increase storage temperature and respiration rate, accelerating nutrient consumption and FCFV decay.

## 3. Novel Methods for Preserving Fresh-Cut Fruits and Vegetables

### 3.1. Physical Methods

#### 3.1.1. Modified Atmosphere Packaging (MAP)

One of the most popular physical methods for preserving FCFVs is modified atmosphere packaging (MAP). This involves altering the gases surrounding a commodity to generate a mix of gases that is distinct from that of air. Pressurized inert gases and oxygen levels above 70 kPa in MAP extend the shelf life of minimally processed fruits and vegetables by adjusting the gas composition within the packaging to slow biological respiration and microbial growth [28].

FCFV respiration consumes oxygen (O_2_) while producing carbon dioxide (CO_2_), reducing O_2_ levels and increasing CO_2_ concentrations in the packaging. Equilibrium occurs when CO_2_ production and O_2_ consumption rates equal those provided through the film. The gas composition at equilibrium is influenced by the product weight, respiration rate, maturity, environmental conditions (temperature and humidity), and the properties of the packaging material, such as perforation density, permeability, film thickness, and surface area [12,29].

MAP provides significant benefits for the preservation of FCFVs. It prolongs shelf life by slowing the respiration rate and delaying ripening, maintaining produce quality during transportation and storage. It also restricts the growth of spoilage microorganisms and maintains the sensory characteristics of fresh-cut produce, including color, texture, and flavor [28]. For baby spinach stored at 4 °C and 7 °C, Brown et al. observed that air packaging and active modified atmosphere packaging (high O_2_ level) had a more pronounced inhibitory effect than a high N_2_ atmosphere [30]. MAP with 5% O_2_, 5% CO_2_, and 90% N_2_ successfully delayed the growth of microorganisms on fresh-cut cucumber kept at 4 °C [31]. Also, packaged fresh-cut broccolis with cellulose-based film efficiently controlled condensation, which maintained O_2_ and CO_2_ levels and inhibited microbial growth [32].

Furthermore, *L. monocytogenes* exhibits variable behavior on fresh-cut produce with MAP at refrigeration temperature. After being exposed to a simulated distribution chain for 7 days under active MAP (3% O_2_/5% CO_2_), *L. monocytogenes* can grow on cucumber and survive on lettuce salad but decreases on bell peppers [33]. Similarly, in a different study, *L. monocytogenes* populations declined on carrots and lettuce kept at 4 °C in two active MAP atmospheres [34]. However, an increase in *L. monocytogenes* on iceberg lettuce packaged under active MAP (4.6–6.2% CO_2_/2.1–4.3% O_2_) and stored at 5 °C and 13 °C was also reported [35]. These reports indicate that using MAP as a single preventative approach does not help stop foodborne disease spread. Thus, further studies should evaluate MAP in concert with other treatments. There are concerns regarding the safety of products using MAP, primarily due to the intended inhibition of spoilage microorganisms, to prolong shelf life, relative to food stored in standard air conditions, potentially facilitating the proliferation of slower-growing pathogenic bacteria.

Recently, a novel form of spontaneous modified atmosphere technology, laser microporous film (LMF) packaging, uses laser perforation to punch small precise circular holes of varying sizes into plastic film to regulate the gas exchange between the film and the environment. Various LMF parameters have been established for horticultural products with distinct respiratory processes and are commonly used in the storage of fruits and vegetables [36,37]. Moreover, MAP combined with edible coatings may enhance the microbiological stability and quality of FCFVs, prolonging their shelf life [38]. To increase the supply of FCFVs and give producers and consumers products of the greatest safety, quality, and shelf life, these methods must be studied in highly perishable fruits and vegetables (i.e., high susceptibility to enzymatic browning, loss of integrity, and microbial spoilage), which have received more attention.

#### 3.1.2. Pressurized Inert Gases

Non-reactive pressurized inert gases such as argon, helium, and nitrogen are used in FCFV preservation to maintain quality and extend shelf life. These gases are excellent barriers to the oxidative processes that reduce food quality [39]. The effect of these gases is based on the removal of oxygen from the packaging environment to prevent oxidative reactions and inhibit aerobic spoilage microorganisms. As oxygen is depleted, the respiration rate of the produce slows, which delays the onset of ripening and senescence [26]. This technique is particularly helpful as it avoids the need for chemical preservatives, aligning with consumer preferences for minimally processed foods. Inert gases are non-reactive and do not impart any odor or off-flavor to the packaged products, maintaining the sensory qualities of fruits and vegetables [40].

Shen et al. studied the effect of pressurized argon and nitrogen treatments, combined with a modified atmosphere, on the qualitative attributes of fresh-cut potatoes during cold storage [38]. For 60 min, fresh-cut potatoes were pressured with argon, nitrogen, or a 1:1 combination of the two gases at 4 MPa. The potatoes were then packed in bags containing 4% oxygen, 2% CO_2_, and 94% N_2_ and kept at 4 °C. Argon treatment delayed the losses of vitamin C, moisture, firmness, and color, significantly reducing the respiration rate and membrane oxidation. The preserved fresh-cut potatoes had a much lower microbial count. The pressurized combined argon/nitrogen treatment was more effective at preserving the quality of fresh-cut potatoes than nitrogen treatment [6]. Thus, pressurized argon or mixed gas treatment with a changed environment protects the quality of fresh-cut potatoes throughout storage.

The retention of quality extends to the phenolic compounds and antioxidants essential for the nutritional and functional quality of fresh produce. Studies have combined inert gases with preservation methods such as MAP or vacuum sealing. Pressurized inert gases combined with MAP enhanced the shelf life of fresh-cut spinach by suppressing the activity of PPO, which is responsible for browning [41].

#### 3.1.3. Electron Beam Irradiation (EBI)

Electron beam irradiation (EBI) is an emerging method for preserving FCFVs, as it inactivates pathogens without affecting sensory qualities such as texture and color. EBI breaks microbial DNA using high-energy electron beams, inactivating harmful pathogens such as *E. coli* and *Salmonella* that can affect minimally processed produce [42]. Low-dose EBI retains firmness, color, and nutritional content, particularly vitamins and antioxidants, better than thermal processes. However, nutrient sensitivity remains a challenge. A study of fresh-cut strawberries reported a 15% loss in vitamin C content, acceptable within the tolerance limits for product quality [43]. This effect is less distinct for stable vitamins, such as B_12_ and fat-soluble vitamins, which undergo minimal changes.

In another study, the total aerobic bacteria counts of cantaloupe slices decreased as the EBI dose increased; all irradiation doses resulted in consistently lower counts for cantaloupe pieces from melons rinsed with chlorine as compared to with water [44]. Fresh-cut packed cantaloupe can withstand up to 1.5 kGy of radiation without deterioration of its qualitative traits. However, EBI is more expensive than other methods, especially its installation, which restricts implementation by smaller producers. No doubt, continuous outreach, plus further refinements in technology and increasing precision of dosage delivery with decreased costs, will increase acceptance by consumers and across the industry [45].

Maintaining a low level of irradiation is essential for preserving fresh fruits and vegetables. According to the US Food and Drug Administration (FDA), fresh fruits and vegetables can receive a maximum radiation dose of 1.0 kGy, except for fresh spinach and lettuce, which may receive radiation up to 4.5 kGy. The safety and feasibility of irradiating all fresh fruits and vegetables at levels up to 4.5 kGy is being studied, even if the maximum dosage for fresh and fresh-cut fruit and vegetables is unclear. The promising findings may drive further study, despite the paucity of the literature on the subject and the rigorous limitations of EBI use on FCFVs [45].

#### 3.1.4. Pulsed Light (PL)

Pulsed light (PL) is a novel preservation technique currently under investigation. Intense, short (1 μs–0.1 s) pulses of broad-spectrum light at wavelengths from ultraviolet to near-infrared (200–1100 nm) are used. PL quickly inactivates bacteria on equipment, surfaces, and packaging materials that come into contact with food. The U.S. FDA has approved this technology for food processing. It is widely accepted that the UV-C component of pulsed light has the greatest bactericidal effects [46]. Unlike UV-C radiation, however, PL destroys microorganisms by various processes, including interconnected photochemical, photothermal, and photophysical effects. PL is a multi-target method that is more efficient than continuous UV radiation at inactivating microorganisms comparatively quickly [47].

Pulsed light prolonged the shelf life of fresh-cut pineapple, with minimal effects on texture and antioxidant capacity [48]. PL treatment prolonged the shelf life of fresh-cut cantaloupe by preserving its physical (firmness, color, and fluid loss) and chemical (pH, titratable acidity, total soluble solids, phenolic content, and ascorbic acid content) properties and tissue structure [49]. When fresh-cut cantaloupe treated with PL was stored at 4 °C, its shelf life was 20 days longer than the control in terms of microbial quality. The extended shelf life results from the reduction in spoilage microorganisms and the preservation of quality attributes. For instance, PL decreased the populations of native mesophilic bacteria without significantly altering the antioxidant capacity, vitamin C, total phenolics, color, and firmness of tomatoes, plums, cauliflowers, strawberries, and sweet peppers [47]. The application of PL (14 J cm^−2^) to fresh-cut avocado reduced mesophilic microorganisms by 1.20 log (CFU g^−1^), inhibited the growth of yeasts and molds, and extended the microbiological shelf life to 15 days at 4 °C [50].

Pulsed light positively influenced the surface decontamination of fresh-cut avocados; although the maximum dosage of 14 J cm^−2^ increased microbial inactivation after processing fresh-cut avocados, the lowest counts were achieved from days 5 to 11 of storage when PL was applied at 3.6 J cm^−2^, significantly reducing microbiological deterioration and extending shelf life by 15 days [50]. According to Ramos-Villarroel et al., 15 or 30 light pulses at 0.4 J cm^−2^ per pulse inactivated *L. innocua* and *E. coli* in avocado cylinders, while reducing ethylene production [51]. Like UV treatment, PL may trigger the production of several beneficial phytochemicals.

The efficacy of PL decontamination appears to be influenced by the composition of the food because food components absorb light. This technique is ineffective when treating proteinaceous or oily foods because these components reduce the efficacy of PL. However, vegetables and fruits that are low in protein and fat may be suitable [46]. The literature on using PL in fruits and vegetables is limited, especially with fresh-cut products.

In addition, the combination of PL with other preservation methods, such as MAP, has shown synergistic effects, further reducing the microbial load while maintaining product quality and shelf life [52]. This combined approach is of interest to the industry for its enhanced efficacy in microbial control and its ability to preserve the safety and sensory qualities of fresh-cut produce.

#### 3.1.5. Ultraviolet Light (UV)

Ultraviolet radiation is categorized by wavelength into UV-A (315–400 nm), UV-B (280–315 nm), and UV-C (100–280 nm). UV-C is non-ionizing radiation, regularly used because of its germicidal effect at a wavelength of 254 nm. By destroying DNA or RNA, such radiation significantly lowers the pathogen load on produce surfaces, halting the deterioration of fresh products and lowering related safety risks. UV-C preserves texture, color, and nutritional value during extended storage by lowering the microbial load and delaying the enzymatic processes that degrade quality [53].

UV-C efficiently decreases microbial populations on fresh and fresh-cut produce while maintaining quality. For example, the color, juice seepage, and overall visual quality of fresh-cut watermelon are maintained while achieving microbial reductions equivalent to those of aqueous sanitizers like chlorine and ozone with a lower dosage of UV-C radiation (13.7 kJ m^−2^ at 254 nm) [54]. *Salmonella* on tomatoes and fresh-cut lettuce was effectively inactivated by water-assisted UV, which involves treating samples with UV light while immersed in agitated water [55]. UV-C treatment avoided surface damage and extended shelf life due to its antimicrobial activity on the *Pseudomonas* species that cause such lesions [56].

UV-C can also potentially mitigate browning in fresh and fresh-cut produce. UV-C treatment (1.5–3 kJ m^−2^ for 5 and 10 min) inactivated the enzymes (PAL, PPO, and POD) involved in browning, which also decreased the amount of soluble quinone without altering the amount of soluble solids or firmness [57]. The effect of UV exposure depends on the dosage, type of fresh produce, duration of treatment, and storage conditions. Without affecting pH or titratable acidity, UV-C also increased the antioxidant capacity of tomatoes and apples during storage by raising their amount of lycopene, total carotenoid, and phenolic compounds [58].

Additional research should assess the detrimental effects that the development and commercial implementation of this technology can have on some produce.

#### 3.1.6. Cold Plasma (CP)

Plasma is an ionized gas comprising photons, free electrons, excited or unexcited atoms, and positive and negative ions and molecules [55]. Cold plasma, also called non-thermal plasma or atmospheric cold plasma, is produced at ambient temperatures and normal atmospheric pressure. CP can inactivate harmful microbes on a variety of surfaces and foods [59].

Fresh and fresh-cut vegetables in sealed packaging can be treated with cold plasma. Under normal pressure, for instance, CP was reported to eliminate *E. coli* from fresh-cut cucumber slices; nevertheless, the treatment did not significantly improve the cucumbers’ physicochemical characteristics and sensory characteristics [60]. Additionally, by preventing oxidative reactions and restricting the growth of microbes, a short CP treatment using a high voltage extended the shelf life of fresh-cut Hami melons without compromising on odor, flavor, and color [61].

Fresh-cut foods often show smaller reductions in dangerous bacteria compared to intact fruit with smooth surfaces. CP treatment significantly decreased the *E. coli* O157:H7 and *S. typhimurium* populations on fresh-cut lettuce by up to 2.8 log (CFU g^−1^), while preserving physicochemical and sensory characteristics [62]. Dielectric barrier atmospheric cold plasma (DACP) did not affect the color, CO_2_ generation, weight, or surface morphology of fresh-cut romaine lettuce during storage and only decreased the *E. coli* O157:H7 and total aerobic microorganisms counts by about 1 log (CFU g^−1^) [63]. *E. coli* O157:H7 populations on packaged romaine lettuce leaves in a five-layer arrangement were decreased by 0.4–0.8 log CFU per piece of lettuce, with no noticeable changes in the respiration rate, weight loss, color, or surface shape [64]. Table 1 summarizes the effects of CP on microbial populations and the quality characteristics of some FCFVs.

CP is an emerging technology that can potentially reduce the risk of pathogen contamination while maintaining the quality of fresh and fresh-cut produce. CP enhanced the antibacterial activity of green tea extract on fresh-cut dragon fruit [65]. However, the majority of studies have been on a laboratory scale, typically treating only a few fresh produce items. The ability to combat human viruses found in fresh fruit will probably be limited when CP is scaled up. The technology requires validation through pilot-scale and commercial trials.

Other limitations of CP technology include the need for the sample to be near the plasma source for some CP applications and lengthy treatment times. Despite these restrictions, plasma has the potential to be integrated into the packaging line of fresh-cut produce processing plants. CP and modified environment packing can preserve fresh-cut product quality during storage after treatment [75].

### 3.2. Chemical Methods

#### 3.2.1. Acidic Electrolyzed Water (AEW)

Acidic electrolyzed water (AEW) is a scientifically substantiated, potent alternative for extending the shelf life and maintaining the quality of FCFVs by minimizing microbial contamination and biochemical degradation. The low pH (2.5–3.5) and high oxidation–reduction potential (>1000 mV) of AEW make it effective against various pathogens, including *L. monocytogenes* and *E. coli*, both important in food spoilage and safety [76]. AEW may also inactivate several harmful and spoilage bacteria species, with minimal adverse effects on humans and on the organoleptic and nutritional quality of foods. It effectively inactivates bacteria on fresh-cut carrots, pears, and apples and fresh, ready-to-eat vegetables and sprouts [77]. The acidic environment of AEW stabilizes cell membranes, reducing cellular leakage, a pivotal contributor to rapid spoilage in fresh produce [78].

AEW treatment also preserves bioactive compounds, notably antioxidants like ascorbic acid (vitamin C) and polyphenols, which are prone to degradation in untreated produce. For example, AEW-treated longan fruit retained 30% more vitamin C over a 14-day storage period than untreated samples. This retention is attributed to oxidative inhibition by AEW, which reduces the internal biochemical stress that accelerates nutrient breakdown [79]. Such nutrient preservation is also important for antioxidant-rich produce, such as strawberries and leafy greens. The ability of slightly acidic electrolyzed water (SAEW), a milder variant of AEW, to balance antimicrobial efficacy with minimal impact on flavor and texture has been evaluated in SAEW-treated cilantro, which had significantly reduced microbial counts without compromising sensory quality, making SAEW a promising option for delicate herbs [80]. Moreover, combining SAEW with mild heat further reduces bacteria and is particularly effective for pathogens resistant to cold treatments [81].

#### 3.2.2. Nanotechnology

Nanotechnology has emerged as a transformative method for preserving FCFVs, mainly by enhancing barrier properties, antimicrobial activity, and moisture retention. Studies have demonstrated the efficacy of nanoscale materials, such as nanoemulsions, edible coatings, and nanocomposites, in slowing microbial growth and oxidative spoilage in fresh produce. One study explored the use of essential oil-based nanoemulsions on fresh-cut fruits, showing that these nanoemulsions substantially delayed microbial growth and preserved the sensory quality of strawberries and apples over prolonged storage periods [20]. The nanoscale droplets in the emulsion form a protective layer, reducing oxidation rates by limiting exposure to air. This also decreases water loss, a common issue with fresh-cut produce, which often suffers from rapid dehydration. By forming a thin stable barrier, nanoemulsions enhance moisture retention, which is critical in the mechanism of nano-coatings for maintaining produce quality [82].

A study showed that packaging fresh-cut melon slices in nanocomposite films made using clay nanoparticles with MAP created an environment that slowed respiration and oxidation; the nanocomposite film significantly improved the gas barrier, reducing the oxygen permeation rate. As a result, the spoilage rate was reduced, and the color and texture were preserved [83]. The surface area of nanoparticles enables better interaction with microbial cells, ensuring thorough microbial inhibition without compromising the sensory properties of the food [84]. The study attributed these results to the intrinsic antimicrobial properties of chitosan, which acts by disrupting microbial cell walls. In addition, Wang et al. evaluated the role of cellulose nanofibrils integrated into biopolymer matrices in maintaining the antioxidant capacity and firmness of apple slices over an extended period [85]. Preserving these bioactive components enhanced the overall nutritious quality of the produce, offering a solution that balances extended shelf life with retained health benefits.

Recent research underscores the efficiency of nanoparticle-loaded washing solutions as a pre-treatment to sanitize and extend shelf life. Nano-encapsulated antimicrobial agents such as carvacrol eliminate surface pathogens effectively and reduce off-flavors, preserving sensory qualities without compromising safety [86].

#### 3.2.3. Ozone

Ozone (O_3_) is a potent antimicrobial agent that targets microorganism cell membranes, causing cell disruption and death. It spontaneously decomposes to non-toxic O_2_ and has high reactivity, penetrability, and aqueous or gaseous application. Ozone treatment positively influences the shelf life of fresh uncut produce, including pears, cucumbers, strawberries, broccoli, grapes, apples, oranges, and raspberries, by reducing microbial populations and facilitating ethylene oxidation [87]. As a gas, ozone permeates surfaces and irregular shapes, enabling uniform coverage on complex surfaces, like fresh-cut produce. Gas application significantly reduces spoilage microbes, particularly in fresh-cut lettuce and other leafy greens [88]. Therefore, it can inactivate a broad spectrum of bacteria, fungi, and viruses without leaving harmful residues on treated produce.

Moreover, another study assessed the efficacy of aqueous ozone (≥3.3 mg min^−1^ flow rate) and ultrasound (40 kHz, 100 W) in inhibiting microbial growth, reducing pesticide residues, and preserving quality during 12-day cold storage (2 °C, 95% RH) on strawberries. The most effective treatment was a 3 min dual dose, which reduced bacterial survival by 98%, removed 98–99% of pesticides, delayed fungal decomposition by 4 days, and minimized weight loss (4.7% vs. 7.7% in controls) [89]. In comparison to untreated fruit, it also improved the retention of nutritional attributes (anthocyanins and ascorbic acid), antioxidant enzyme activity, and visual quality, thereby extending the marketable storage life by six days. The combination of ultrasound and ozone resulted in improved disinfection and quality retention, without the presence of chemical residues [89].

In addition to microbial control, ozone treatment slows enzymatic browning and oxidation processes that typically reduce the visual appeal and nutritional quality of fresh-cut produce. Notably, it prevents color degradation and texture loss in sliced apples and strawberries, extending shelf life by maintaining appearance and firmness [90]. For practical applications, gaseous ozone treatments can easily be added to processing facilities with minimal infrastructure changes [90]. Low ozone concentrations (1–5 ppm) can deactivate pathogens like *E. coli* and *L. monocytogenes* on fresh-cut lettuce without compromising quality [91].

It is important to consider the adverse health consequences of exposure to elevated ozone concentrations for prolonged durations. Increased ozone exposure and concentrations are linked to irritation of the eyes, throat, nose, and respiratory system, lung damage, chronic respiratory illness, edema, and bleeding. The U.S. Federal Occupational Safety and Health Administration established threshold limits of 0.1 ppm for long-term ozone exposure (8 h) and 0.3 ppm for short-term exposure (15 min) in the workplace [91].

#### 3.2.4. Chlorine Dioxide

Chlorine dioxide (ClO_2_) is an effective antimicrobial agent in the food industry, especially in relation to FCFVs. Due to its oxidative nature, ClO_2_ disrupts microbial cell walls, leading to cell death, effectively reducing spoilage and extending freshness life [88]. Compared to traditional sanitizers, ClO_2_ can achieve this at lower concentrations, reducing the chemical residuals on food surfaces, as confirmed in studies in which ClO_2_ significantly reduced microbial loads in various fresh-cut produce commodities, enhancing safety and freshness. The oxidation capacity of ClO_2_ is 2.5 times that of chlorine [92].

ClO_2_ does not react with ammonia or substances that contain nitrogen to form dangerous chloramine compounds; it has antibacterial properties and it is recognized for use in fruit and vegetables. ClO_2_ was effective against *E. coli* O157:H7-inoculated fresh-cut lettuce and baby carrots [93]. ClO_2_ effectively deactivates *L. monocytogenes* and *S. typhimurium* [94]. It can be applied to various produce types, but its action and scope depend on the type of fruit or vegetable, microbial vulnerability, and storage conditions. Berries, especially strawberries, are very perishable fruit due to microbial spoilage, and ClO_2_ gas treatment can sanitize them without excess moisture. Leafy greens like lettuce and spinach have also shown positive results with ClO_2_ treatment, which minimizes microbial contamination without affecting the texture or color of the tender leaves. Thicker-skinned fruits, like apples and citrus fruits, may rely on the antimicrobial action of ClO_2_ to control fungal growth and reduce superficial spoilage [93,94].

The primary limitation of ClO_2_ is its permitted use in produce at 3 ppm. Research has revealed that prolonged exposure to ClO_2_ or high concentrations can alter produce color, taste, or other sensory parameters, particularly in delicate leafy greens and strawberries. Treatment doses and the exposure duration must be examined carefully to prevent oxidative damage to the product. The US Code of Federal Regulations requires that, following ClO_2_ treatment, the product be washed with drinkable water [95]. Nevertheless, studies confirm that these residues are much lower than those of other traditional chlorine-based treatments and within the limits of food consumption safety.

Combining UV-C light with ClO_2_ treatment is promising, as they work synergistically to eliminate microbial contaminants, as evident in fresh-cut products [57]. With further research and development, ClO_2_, especially when combined with other methods, has great promise for improving safety and increasing the shelf life of FCFVs to meet consumer requirements and market demands.

#### 3.2.5. Edible Coatings

Using edible coatings as a packaging method increases the shelf life of FCFVs [22]. It affects the flow of water, which in turn reduces the amount of moisture lost from the fruit surface. It can also affect the surrounding microenvironment by acting as a barrier to gas interchange (Figure 3).

After harvesting, a substantial amount of the natural waxy covering is removed from the fresh product by washing it to remove the dirt and soil residues and enhance its sensory appeal. If there is no natural waxy coating, fresh vegetables may shrivel, wilt, and have an unfavorable texture due to excessive moisture loss [1]. An edible coating changes the environment within the fruit and prevents moisture, gases, and solutes from moving through the thin coating membrane. Coatings can be applied by brushing, spraying, or dipping. The ideal coating regulates anaerobiosis and withstands decay without compromising the quality of the finished fruit [96]. To improve the microbiological quality of edible material and reduce fatty acid oxidation, which ultimately reduces enzymatic browning and texture changes during the shelf life, antibacterial, antifungal, and antioxidant materials such as ascorbic acid, L-cysteine, and citric acid can be used [11].

The use of polysaccharide- and protein-based edible coatings for fresh-cut items has been discussed in recent publications [3,5]. Gas barrier properties change the atmosphere and increase shelf life without creating highly anaerobic conditions, while polysaccharide- and protein-based coatings can serve only as a slight moisture barrier because of their hydrophilic nature [1,20]. Fruits and vegetables subjected to minimal processing have been efficiently coated with polysaccharides and proteins, including various natural gums from diverse sources, starches, dextrin, pectin, cellulose and its derivatives, chitosan, alginate, carrageenan, gellan, gelatin, casein, soy protein, and zein [20]. Table 2 summarizes the application of edible coatings in some FCFVs.

The use of emulsion-based edible coatings was recently recognized as a way to prolong the shelf life of fresh-cut fruits and vegetables. Edible coatings based on emulsions can be made using a variety of essential oils, vegetable and animal oils, vegetable and animal waxes (carnauba wax and beeswax), emulsifiers, and water [82,106]. Therefore, emulsion-based coatings open a new avenue for connecting the properties of hydrophilic and lipophilic functional compounds [20].

To improve chemical stability and viability while protecting FCFVs from heat, moisture, and other adverse environments, they are now being micro- and nano-encapsulated with edible coatings to help control their release under specific conditions. Although materials from different sources can be used, alginate is most often used for encapsulation. Marine oils (omega-3 fatty acids), probiotics, prebiotics, and enzymes are good functional ingredients for encapsulation [97].

Further research is required to understand how active compounds and coating materials interact when producing novel edible films and coating applications. The mechanical, sensory, and functional qualities of edible films and coatings may be significantly affected by adding active substances (antimicrobials, nutrients, and antioxidants). Research on this is limited, requiring further information to advance coating applications with enhanced functionality and outstanding sensory performance.

### 3.3. Biopreservation Methods

#### 3.3.1. Bacteriocins

Gram-positive and -negative bacteria produce peptides known as bacteriocins with bactericidal or bacteriostatic properties [112]. Larger bacteriocins are peptides produced by ribosomes; their structures and biological characteristics vary. Lactic acid bacteria (LAB) produce bacteriocins, which have a relatively broad spectrum of inhibitory effects. The food industry can produce foods with rich organoleptic and nutritional qualities, and naturally preserved foods using bacteriocins, to reduce the demand for chemical additives and the severity of heat treatments. In addition to developing “novel” food items (such as less acidic or salty), this might help meet growing customer expectations for safe, fresh-tasting, ready-to-eat foods that have undergone minimal processing [113].

As potential natural food preservatives against pathogenic germs and spoilage, bacteriocins have long piqued the curiosity of the food industry. LAB produce nisin, pediocin, and other bacteriocins that have drawn attention for their effects on food production and human health, and their ability to replace chemical preservatives whose safety is constantly being questioned. This provides a rationale for the growing trend in LAB applications in the food business [114].

Research is currently exploring the application of bacteriocins for preserving fruits and vegetables, given their inherent preservative properties. The FDA has approved nisin, Micocin, and pediocin PA-1/AcH for use as food additives. Since they function as natural antimicrobials and substitutes for conventional food preservatives, research has assessed their use in fruits and vegetables [112]. Bacteriocins are being indirectly introduced into foods, allowing producer strains to be injected into fresh produce and manufacture bacteriocins in situ. As food additives, bacteriocins can also be directly applied to FCFVs to improve their microbiological safety [115].

#### 3.3.2. Bioprotective Microorganisms

Bioprotective microorganisms, mainly LAB and other antagonistic microbial strains, are effective preservatives in FCFVs. These exert their preservative action on spoilage and pathogenic microorganisms via competitive exclusion and the production of antimicrobial compounds, leading to increased safety and shelf life. Previous studies illustrate that this food biopreservation method promotes shelf-life extension of foods, maintaining both safety and quality [116]. LAB and other inhibitory microorganisms prevent pathogens and spoilage via mechanisms such as competitive exclusion, acidification, and bacteriocin formation.

Bacteriocins include peptides that interfere with sensitive target cell cellular integrity, inhibiting further development on fresh produce surfaces. Organic acid production via lactic or acetic acid decreases the pH, presenting an unfriendly environment for pathogens that cause spoilage [117]. This markedly reduces the microbial load, maintaining the safety and quality of these perishable commodities.

For instance, the application of *Lactobacillus* spp. and *Pediococcus acidilactici* significantly reduced the *Listeria monocytogenes* by 1.2–1.6 log (CFU g^−1^) in fresh-cut lettuce after treatment [118]. Another study identified several LAB strains, such as *Leuconostoc mesenteroides*, that have already been shown to prevent *L. monocytogenes* in fresh produce and can potentially be applied for the biopreservation of ready-to-eat salads and similar fresh-cut commodities [119]. In this regard, hydrogen peroxide, diacetyl, and reuterin are the metabolic by-products of LAB, providing an effective protective barrier against numerous spoilage organisms.

Similarly, bioprotective microorganisms are a natural source of alternative chemical preservatives that satisfy the need for minimally processed food. Biopreservation confers safety and quality on goods, similar to traditional preservatives, but presents fewer health hazards and less environmental concern. It allows the produce industry to reduce reliance on chemical treatments, which appeals to the organic and natural foods market [116].

#### 3.3.3. Bacteriophages

A bacteriophage or phage is a virus that infects and kills bacteria [120]. These are among the most plentiful, self-replicating biomaterials on Earth. There may be as many as 10^30^–10^31^ phage particles in soil, which typically outnumber the bacterial host by more than a logarithmic fold [121]. Bacteriophages are important biocontrol agents, targeting various bacterial pathogens found in food, and can significantly reduce specific common bacterial contaminants in fresh and minimally processed foods, including *Campylobacter*, *E. coli*, and *Salmonella* [122].

Oliveira et al. reported that the Listex^TM^ P100 is effective in biocontrolling *L. monocytogenes* growth on pear and melon slices, and in their juices during storage at 10 °C, while the phage did not exhibit any impact on apple liquids or slices [123]. They also found that the acidity of the lytic environment influences phage–host interactions, as validated when the degree of *L. monocytogenes* decreased in three distinct fresh product types (apples, pears, and melons) with varying pH levels (5.92, 4.91, and 3.76) was used to assess the extent to which phage treatment worked. Therefore, the phage’s efficacy may be improved by incorporating a combination of other technologies for low-pH meals (such as juices or fresh-cut).

Moreover, another study revealed that the amount of diffusion in the test medium was correlated with the practical effectiveness of the bacteriophage; diffusion is greater in liquid medium than in solid matrix, where phage immobilization is likely. This was supported by findings that the phage-treated fresh melon juice had an 8.00 log CFU mL^−1^
*L. monocytogenes* population decrease compared to 1.50 log CFU mL^−1^ in fresh-cut melon [124]. Nevertheless, further study is necessary to enhance the antibacterial efficacy of bacteriophages and minimize the contact duration. In the future, high-tech and efficient control of bacteriophages is expected by researchers to be a major component of future countermeasures against antibiotic-resistant bacteria. Still, the future looks a little distant to fit phage uses in fresh produce operations.

## 4. Conclusions

This review addresses critical challenges regarding the loss of quality in fresh-cut fruit and vegetables during processing and storage that lead to spoilage and contamination. Understanding the loss of quality and sources of microbial contamination guides the development and choice of effective preservation techniques that can minimize spoiling and foodborne pathogens (Figure 4). Processing methods affect the quality of fresh-cut produce, emphasizing the need for innovative strategies that maintain or enhance sensory and nutritional properties. Novel methods such as cold plasma, edible coating, and natural preservatives have shown promise in inhibiting microbial growth while preserving texture, flavor, and shelf life. However, further research is needed to optimize these techniques, particularly regarding microbial resistance, and their applicability to various fresh-cut produce types. Future studies should enhance the nutrient value and organoleptic functionality of FCFVs with combinations of these advanced technologies. Although advances in traditional and novel preservation technologies are notable, concerns remain about their long-term effects on food quality, consumer acceptance, and the environment. Continued research is crucial to address these problems and support a healthier marketplace.

## Figures and Tables

**Figure 1 foods-14-02769-f001:**
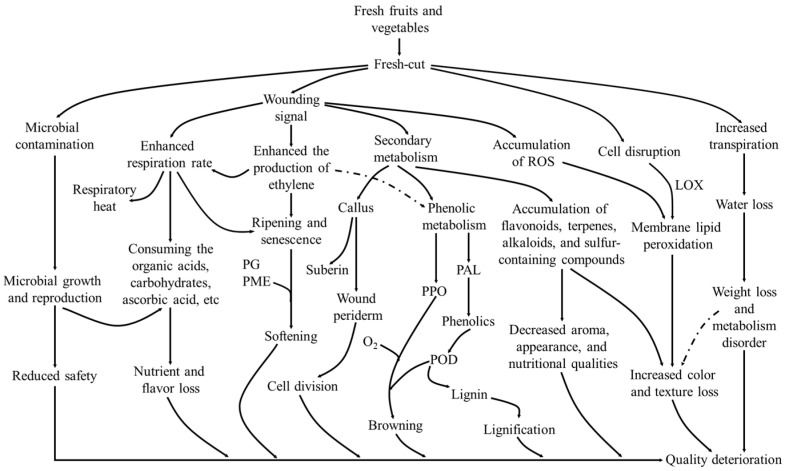
The reasons for quality deterioration of fresh-cut fruits and vegetables [7].

**Figure 2 foods-14-02769-f002:**
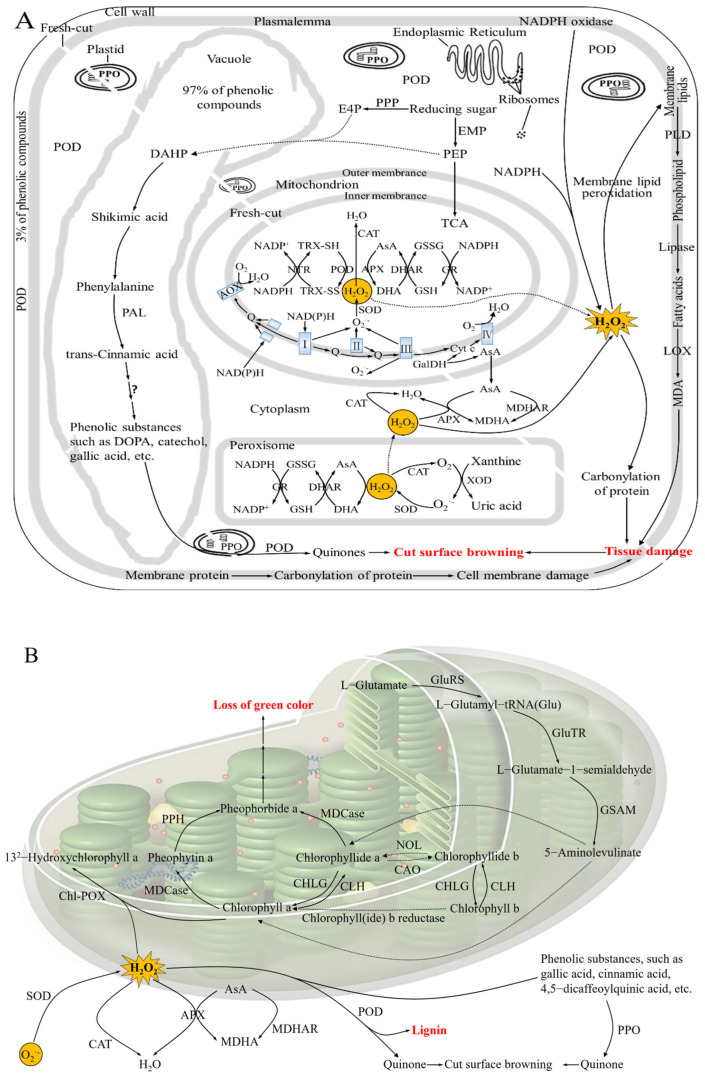
The possible mechanism of cut surface browning (**A**) and green color loss (**B**) in fresh-cut fruits and vegetables.

**Figure 3 foods-14-02769-f003:**
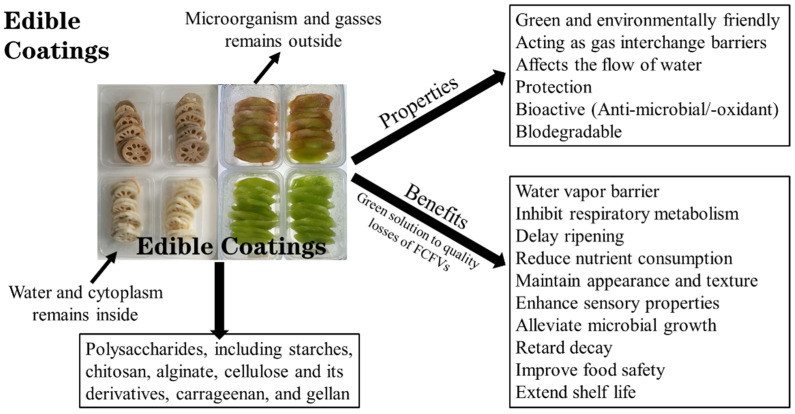
The possible role of edible coating to prevent the quality losses in fresh-cut fruits and vegetables.

**Figure 4 foods-14-02769-f004:**
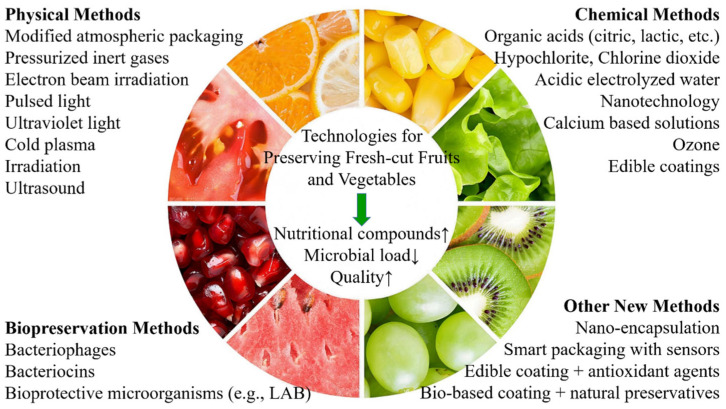
Summary of the technologies for preserving fresh-cut fruits and vegetables.

**Table 1 foods-14-02769-t001:** Application of cold plasma on fresh-cut fruits and vegetables.

Treatment Conditions	Fruits and Vegetables	Effect on FCFVs	References
Water activated by CP (a plasma treatment performed on 2.5 L of distilled water at 2.6 kV/20 min).	Mango	Inhibited the growth and reproduction of microbes, retarded the membrane lipid peroxidation, and delayed quality loss such as color, weight, firmness, titratable acidity, and soluble solids contents.	[2]
Plasma-activated water (0.5 L of water was subjected to plasma at 51.7 W, 14.4 kHz, 8 kV/61.5 min).	Celery	The inoculation of *L. monocytogenes* and *E. coli* O157:H7 on celery resulted in inactivation levels of 1.1 ± 0.1 and 1.2 ± 0.1 log (CFU g^−1^), respectively.	[55]
CP at 40 kV/90 s.	Cantaloupe	Reduced the development and reproduction of mold and bacteria, enhanced firmness, quality, and sensory characteristics; aromas with floral and fruity notes were significantly improved.	[59]
Dielectric-barrier discharge (DBD) cold atmospheric plasma at 23 kV/2.5 or 8.5 min.	Apple	Eliminated the total viable count and patulin, reduced the PPO activity and total phenolic content, and increased the quality and lifespan.	[65]
In-package CP product; storage time: 1, 4, and 7 days; exposure time: 0, 1, and 3 min.	Apple	Improved the fruit’s quality and maximum antioxidant concentration and decreased PPO activity, but not enough to maintain the product’s antioxidant qualities.	[66]
DBD CP at 75 kV/3 min.	Mango	Prevented microbiological growth and preserved physicochemical characteristics, delaying the loss of organoleptic and nutritional properties.	[67]
DBD CP at 45 kV/1 min.	Strawberries	Maintained the textural qualities, inhibited microbiological growth, and increased the flavonoids, anthocyanins, and phenolic compounds.	[68]
CP at 60 kV/5 min.	Pitaya	Enhanced antioxidant activity, accelerated phenolic accumulation, improved energy status, promoted the consumption of primary sugars, and inhibited the growth of aerobic microorganisms.	[69]
High voltage cold atmospheric plasma (HVCAP) for 2 and 5 min.	Baby spinach leaves	Eliminated microbiota by 2.6 and 3.5 log CFU per sample, respectively, until 7 days in the refrigerator. The 5 min indirect, 80 kV HVCAP treatment did not affect the leaves’ moisture content, color, or texture.	[70]
CP at 50 kV/30 s.	Bamboo shoots	Exhibited higher firmness and reduced yellowing. Six key flavoring chemicals were identified (odor activity value (OAV) > 1), reduced alkenes and alcohols, while keeping ester, aldehyde, and ketone flavor compounds, and increased microbiological diversity.	[71]
CP at 60 kV/5 min.	Carrot	Reduced microbiological development and color changes, enhanced the accumulation of γ-aminobutyric acid, and improved the energy status and reducing power.	[72]
Dielectric-barrier discharge cold atmospheric plasma at 42.54 ± 2.58 W, 23 kHz, 6 kV/15, 30, or 60 min.	Iceberg lettuce	Inhibited PPO and POD activity, reduced chlorophylls, polyphenols, ascorbic acid contents, and antioxidant activity, decreased mesophilic and psychrotrophic bacteria, while it had no effect on yeast.	[73]
2.45 GHz, 1.2 kW for 10 min of plasma-processed air.	Potato and apple	In fresh sliced apple and potato tissue, PPO and POD activity decreased, the pH of the tissue surface decreased, and cell integrity and dry matter content remained unchanged.	[74]

**Table 2 foods-14-02769-t002:** Application of edible coating in fresh-cut fruits and vegetables.

Edible Coating Material	Fruits and Vegetables	Additives	Effect on FCFVs	References
Chitosan and locust bean gum (LBG)	Carrot	Glycerol and potassium sorbate	In Dordogne, chitosan reduced browning, improved functional and microbiological properties; however, in Purple Sun, its poor O_2_/CO_2_ permeability led to browning and hastened degradation; LBG induced deterioration in microbiological quality in both cultivars.	[1]
Sodium alginate	Lotus root	L-cysteine and citric acid	Inhibited the browning and microbial proliferation, maintained the quality, and extended the lifespan of lotus root slices.	[11]
Aegle marmelose fruit shell polysaccharide (AMFSP)	Apple	Glycerol and glacial acetic acid	Shelf life increased by 1% AMFSP coating, maintaining nutrient content and sensory appeal, reducing oxidative degradation, microbial growth, moisture loss, and enzymatic browning.	[20]
Sodium carboxymethyl cellulose	Asparagus lettuce	Ascorbic acid and L-cysteine	Inhibited browning and green color loss, reduced oxidative stress, sustained the quality, and enhanced the lifespan of lettuce slices.	[22]
Nanoemulsion-based alginate	Papaya	Oregano essential oil	Inhibited weight loss, moisture evaporation, microbial growth, respiration rate, color stability, and soluble solids, better antimicrobial efficacy, slight sensory changes, and prolonged shelf life.	[38]
Gum Arabic	Capsicum	Aqueous extracts of *Syzygium aqueum*, *Tasmannia lanceolata*, and *Diploglottis bracteata*	Inhibited microbial development and enhanced sensory attributes; however, it was insufficient in maintaining the retained moisture and firmness of fresh-cut capsicum.	[96]
Sodium alginate (SA)	Apple	Clove essential oil (CEO)	Decreased oxidative damage, moisture loss, and microbial growth, the shelf life was extended to 14 days at 4 °C; 1% CEO nanoemulsion had a more obvious effect for preventing quality loss.	[97]
Sodium carboxymethyl cellulose	Apple	Glycerol and zein	Prolonged shelf life, maintained probiotic viability, improved safety, sustained the microbial quality, and lowered the browning index.	[98]
Sodium alginate	Cantaloupe	Glycerine and thyme oil	Reduced weight loss, prevented *S. aureus*, *S. Typhimurium*, *E. coli* O157:H7, and *L. monocytogenes* growth, and retained color and flavor.	[99]
Chitosan	Guava	Chitosan nanoparticles and citric acid	Controlled postharvest fungal infections, preserved color, physicochemical and sensory qualities, reduced weight loss, delayed ripening, and maintained quality.	[100]
Chitosan	Melon	Ag–chitosan nanocomposites	Reduced respiration rate, prevented softening, maintained a relatively low translucency and high vitamin C content, sensory scored, no significant difference in color, pH, soluble solids, sucrose, glucose, fructose, titratable acidity, and citric and malic acid contents.	[101]
Almond gum from (*prusnus domestica*)	Pineapple	Glycerol monostearate	Extended shelf life compared to synthetic tragacanth gum, delayed color changes, weight loss, and microbial growth, maintained titratable acidity, ascorbic acid, firmness, and total soluble solids.	[102]
Chitosan nanoparticles	Bell pepper	Sodium tripolyphosphate pentabasic	Maintained the fresh-cut bell pepper for 12 days at 5 °C without loss of weight and sensory quality.	[103]
Chitosan, alginate, carboxymethyl cellulose, etc.	Cucumber	—	Maintained the physicochemical and sensory properties, reduced the microbe load, enhanced longevity up to 12 days at 5–7 °C.	[104]
Cactus *Opuntia dillenii* polysaccharide	Potato	—	Delayed browning, microbial proliferation, and respiration rate, reduced sugar accumulation and weight loss of fresh-cut potatoes stored at 5 °C for 5 days.	[105]
Low-molecular-weight chitosan	Red bell pepper	Calcium chloride and tea tree oil nanoemulsion	Retained texture, sensory quality, and overall integrity, suppressed *L. monocytogenes*, *S. enterica*, fungi, and microbial colonization of fresh-cut red bell peppers stored at 4 °C for 18 days.	[106]
Soy protein	Cantaloupe	Glutathione and cinnamon essential oil	Inhibited the growth of *E. coli* and *S. aureus*, reduced mass and firmness loss, maintained the ascorbic acid, total soluble solids, and titratable acidity contents, and extended the shelf life to 10 days at 4 °C.	[107]
Soy protein	Leaf lettuce	Thymol	Prevented microbial development, reduced mass losses, maintained ascorbic acid and chlorophyll levels, retained lettuce characteristic aroma and texture for 6 days at 4 °C.	[108]
High methoxyl pectin and whey protein isolate	Apple	—	Reduced total colony count, inhibited decay of fresh-cut apple stored at 5 °C for 6 days.	[109]
Whey protein isolate	Apple	Montmorillonite and citric acid	Maintained the color characteristics, reduced the loss of acidity, soluble solids, and water activity, inhibited PPO and POD activity.	[110]
Whitemouth croaker protein isolate	Papaya	Organo-clay	Reduced microbial growth and loss of weight, firmness, lightness, and pH of fresh-cut Formosa papaya stored at 5 °C for 12 days.	[111]

Note: —, no additives.

## Data Availability

No data was used for the research described in the article.

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
