# Peer review of "Recent Advances in Technologies for Preserving Fresh-Cut Fruits and Vegetables"

_foods, 2025, doi:10.3390/foods14162769_

Round 1
Reviewer 1 Report
Comments and Suggestions for Authors
Reviewer Report
Recent Advances in Technologies for Preserving Fresh-cut Fruits and Vegetables
The manuscript presents a comprehensive and timely review of preservation technologies for fresh-cut fruits and vegetables, covering a wide range of physical, chemical, biological strategies. The topic is highly relevant to the current needs of both academic research and the food industry.
I found the manuscript to be generally well written, with clear language, a logical progression of ideas, and a commendable depth of discussion across most sections. The authors have done an excellent job in summarizing the literature and presenting the information in a way that will be valuable to researchers, students, and professionals working in this field. The figures, are conceptually informative and enhance the overall clarity of the text. The manuscript has significant educational value and demonstrates a solid grasp of the subject matter.
That said, there are several issues that need to be addressed. My detailed comments can be found below;
- Abstract can be improved, it is repetitive in some parts , like these two sentences that say similar things “This paper presents the potential of numerous novel physical, chemical, and biopreservation methods for reducing mi- crobial growth, to preserve fresh-cut produce, while maintaining product quality” and “This review assesses preservation technologies, their commercialization potential, and future perspectives. “ The ending of the abstract is not conclusive, is not suitable as an abstract ending, it should be rewritten.
- Im not sure about pressurized inert gases to be used as a separate category from MAP. When you modify the gas composition in the package the process is caled MAP so I thing it should be discussed under MAP subheading and not a separate one.
- Future Perspective - Either totally remove future prospects from the study or please write a better more detailed future perspectives section. This is too generic. Future perspective should give information on the growing trends while also using your own educated opinion on the subject to estimate what the future holds. It should also critically discussed subjects to gather and use this foundation to build a roadmap for the future.
Below are more minor and specific corrections:
Line 36 - “barriers to the appeal of FCFVs” its not clear what is meant here, please rewrite this sentence.
Line 46 – “biopreservative” does the author mean biological? “chemical, physical and biopreservatives” do not go well, maybe use the term biological?
Line 63 – 65 - Fix the sentence please, it start with “to…” and ends with “to…”
Line 66 – Rewrite the sentence please, it sounds off (or remove “involved”).
Line 99 – 100 - very abrupt transition, please write a better introduction, the start to give examples.
Line 201 – 203 – This sentence should be supported with more study findings.
Line 416 - There is a problem with numbering of this title, it went from 3.6 to 3.2
Line 440 – 443 - Nanotechnology as a title is a too vague and general term, it is form of technology rather then a specific method, and the introduction sentence make it does not sound clear or scientifically qaccurate. Please use a more define title likes “nanoparcile coatings” “nanoemulsion based layers” or smt similar to these that is both definitive and less generalized. Also adjust the intro sentence accordingly.
Line 525 – 529 - This sentence requires multiple references to support.
Line 564 - Use multiple research based references to support please.
Line 584 - This title is repeated, ther is already a chemical methods with the number 3.2. i think it should be biological methods.
Line 665 - use “extend shelf life” instead of “keep for longer”

Author Response
The manuscript presents a comprehensive and timely review of preservation technologies for fresh-cut fruits and vegetables, covering a wide range of physical, chemical, biological strategies. The topic is highly relevant to the current needs of both academic research and the food industry.
I found the manuscript to be generally well written, with clear language, a logical progression of ideas, and a commendable depth of discussion across most sections. The authors have done an excellent job in summarizing the literature and presenting the information in a way that will be valuable to researchers, students, and professionals working in this field. The figures, are conceptually informative and enhance the overall clarity of the text. The manuscript has significant educational value and demonstrates a solid grasp of the subject matter.
RESPONSE: Thank you for your comments and sincere guidance.
That said, there are several issues that need to be addressed. My detailed comments can be found below;
- Abstract can be improved, it is repetitive in some parts , like these two sentences that say similar things “This paper presents the potential of numerous novel physical, chemical, and biopreservation methods for reducing microbial growth, to preserve fresh-cut produce, while maintaining product quality” and “This review assesses preservation technologies, their commercialization potential, and future perspectives.” The ending of the abstract is not conclusive, is not suitable as an abstract ending, it should be rewritten.
RESPONSE: Thank you for your valuable suggestion. We have revised the abstract according to your suggestions.
Line 19-24: This paper describes the reasons which causes the decline of quality of fresh-cut produce, and presents the potential of numerous novel physical, chemical, and biopreservation methods for reducing microbial growth, to preserve fresh-cut produce, while maintaining product quality. The purpose of this paper is to provide a level of understanding that can be used to underpin future research directions in order to resolve existing issues in this field.
- I’m not sure about pressurized inert gases to be used as a separate category from MAP. When you modify the gas composition in the package the process is called MAP, so I think it should be discussed under MAP subheading and not a separate one.
RESPONSE: We appreciate your observation regarding pressurized inert gases and MAP.
Because the pressurized inert gases belongs to high-pressure preservation technology. We are discussing this technology separately is to emphasize the unique role in fresh-cut preservation.
- Future Perspective - Either totally remove future prospects from the study or please write a better more detailed future perspectives section. This is too generic. Future perspective should give information on the growing trends while also using your own educated opinion on the subject to estimate what the future holds. It should also critically discussed subjects to gather and use this foundation to build a roadmap for the future.
RESPONSE: Thank you for your excellent suggestion.
We have removed this section.
Below are more minor and specific corrections:
Line 36 - “barriers to the appeal of FCFVs” its not clear what is meant here, please rewrite this sentence.
RESPONSE: Thank you for your excellent suggestion.
Line 35-37: We have rewritten the sentence to bring clarity in sentence.
Line 46 – “biopreservative” does the author mean biological? “chemical, physical and biopreservatives” do not go well, maybe use the term biological?
RESPONSE: We appreciate the reviewer’s valuable observation.
Line 46: As suggested, we have replaced “biopreservatives” with “biological preservation methods” to maintain consistency with the terms “chemical” and “physical.”
Line 63 – 65 - Fix the sentence please, it start with “to…” and ends with “to…”
RESPONSE: We appreciate your valuable observation.
Line 63-65: We have revised the sentence, now “Various strategies are employed to enhance the sensory quality and extend the shelf life of FCFVs by slowing their deterioration rate.”
Line 66 – Rewrite the sentence please, it sounds off (or remove “involved”).
RESPONSE: Thank you for highlighting the need for clarity.
Line 66: We have remove “involved”.
Line 99 – 100 - very abrupt transition, please write a better introduction, the start to give examples.
RESPONSE: We appreciate your valuable observation.
Line 99-103: We have improved this section with better introduction and solid example.
Line 201 – 203 – This sentence should be supported with more study findings.
RESPONSE: We thank the reviewer for this suggestion.
We have provided some references to support.
Line 416 - There is a problem with numbering of this title, it went from 3.6 to 3.2
RESPONSE: We thank the reviewer for this suggestion.
Previously, it was mistakenly irregular numbering. In revised manuscript, we corrected the numbering (3.1 with sections from 3.1.1 to 3.1.6), and Chemical methods 3.2.
Line 440 – 443 - Nanotechnology as a title is a too vague and general term, it is form of technology rather then a specific method, and the introduction sentence make it does not sound clear or scientifically qaccurate. Please use a more define title likes “nanoparcile coatings” “nanoemulsion based layers” or smt similar to these that is both definitive and less generalized. Also adjust the intro sentence accordingly.
RESPONSE: We appreciate the reviewer’s feedback regarding the title and introduction.
In this section, we have clarified the scope by specifying the different nanotechnology-based approaches (e.g., nanoparticle coatings, nanoemulsions, and nanocomposites) used in fresh-cut produce preservation. We hope these adjustments address the reviewer’s concerns while maintaining the comprehensive coverage of nanotechnology applications in this field.
Line 525 – 529 - This sentence requires multiple references to support.
RESPONSE: We thank the reviewer for this suggestion.
We have added some relevant references to support the information.
Line 564 - Use multiple research based references to support please.
RESPONSE: We thank the reviewer for this suggestion.
We have added some relevant references to support the information.
Line 584 - This title is repeated, there is already a chemical methods with the number 3.2. i think it should be biological methods.
RESPONSE: We thank the reviewer for correction.
Line 584: Yes, its “biopreservation/biological methods” instead of Chemical methods.
Line 665 - use “extend shelf life” instead of “keep for longer”.
RESPONSE: Thank you for your comments and sincere guidance.
Line 665: We have replaced the word according to your suggestion.

Reviewer 2 Report
Comments and Suggestions for Authors
Dear author,
Thanks for your good review. My comments and suggestions are in the attached file.
Best regards

Author Response
Reviewer #2:
Line 60: Research or review?
RESPONSE: Thank you for your comments and sincere guidance.
It’s “current research”.
Line 76: Shelf life in some fresh cut fruits are more than 3 days at cold storage. Please delete 1-3 days.
RESPONSE: Thank you for your comments.
Line We have delete 1-3 days.
Line 121, 123:
RESPONSE: We have deleted the space between numbers and units.
Line 228, 230: E. coli dose not significant effect on fruit and vegetables decay. So, add some recent studies that show MAP influenced on decay including evaluation of fungi.
RESPONSE: Thank you for your comments.
Line 227-230: We have deleted E. coli studies and have added relevant recent studies to improve MAP section.
Line 249-250: Please add related references.
RESPONSE: Thanks for your valuable comment.
Line 249-250: We have added the reference to support this information.
Line 286: Write E. coli and Salmonella in italic form.
RESPONSE: Thank you for your comment.
We have change into italic form; E. coli and Salmonella.
Line 320-321: Add reference here.
RESPONSE: Thank you for your note.
We have added the reference to support this information.
https://doi.org/10.1111/1750-3841.17255.
Line 363: Italic the word “Salmonella”.
RESPONSE: Thank you for your comment.
We have changed into italic form.
Line 384-389: Please use these two references instead of reference number 61.
Sun, Y.; Zhang, Z.; Wang, S. Study on the bactericidal mechanism of atmospheric-pressure low-temperature plasma against escherichia coli and its application in fresh-cut cucumbers. Molecules 2018, 23, 975. [Google Scholar] [CrossRef]
Tappi, S.; Gozzi, G.; Vannini, L.; Berardinelli, A.; Romani, R.L.; Pietro, R. Cold plasma treatment for fresh-cut melon stabilization. Innov. Food Sci. Emerg. 2016, 33, 225–233. [Google Scholar] [CrossRef]
RESPONSE: Thank you for your valuable suggestions.
We have replaced the references in relevant section.
Line 392-394: reference?
RESPONSE: Thank you for your comment.
We have added the relevant reference.
https://doi.org/10.1016/j.lwt.2023.115384.
Line 412: Notwithstanding
RESPONSE: We have change “Notwithstanding” to “Despite”.
Line 454: Delete it as you mentioned it earlier.
RESPONSE: Thank you for correction.
Line 454: We have deleted the modified atmosphere packaging.
Line 533-534: add related reference.
RESPONSE: Thank you for your correction
We have added the reference.
Line 538: Please mentioned examples of other sources of edible coatings like proteins and lipids.
RESPONSE: Thank you for sincere guidance.
We have added the different sources of edible coatings like proteins and lipids in detail.
Line 565: Add examples of some lipids and proteins EC.
RESPONSE: Thank you for your valuable note.
We have added some examples of both type of EC in table 2.
Line 565: Please shorten long sentences like this in tables.
RESPONSE: Thank you for your comments and sincere guidance.
We have revised and short the sentence as much as possible.
Line 670: correct spelling here.
RESPONSE: Thank you for your comments and sincere guidance.
We have corrected the spelling (compounds).

Reviewer 3 Report
Comments and Suggestions for Authors
Please find the attached file

Author Response
This review addresses critical challenges regarding the loss of quality in fresh-cut fruit and vegetables during processing and storage that lead to spoilage and contamination. Understanding the loss of quality and sources of microbial contamination guides the development and choice of effective preservation techniques that can minimize spoiling and foodborne pathogens.
RESPONSE: Thank you for your comments and sincere guidance.
The work is well-written and comprehensive, covering many new and innovative techniques. However, ultrasound is not given much space, and is only mentioned between lines 482 and 489. Also, pulsed electromagnetic fields, a modern, non-thermal technique that has been widely used recently, are not mentioned. Therefore, it is preferable to include both techniques in the manuscript separately, given their importance in extending the shelf life of food products.
RESPONSE: We sincerely appreciate your valuable suggestion.
We agree with you viewpoint. Ultrasound and pulsed electromagnetic are modern non-thermal technique that have been widely used recently. However, both of two techniques have certain limitations.
Ultrasound is generally used in the initial cleaning process of fresh-cut fruits and vegetables processing, and is only used as auxiliary means. Despite ultrasound treatment can reduce the microbial loads, it has notable limitations. Its standalone microbial reduction is often limited to approximately 1 log CFU/g, making it less effective than traditional sanitizers. Furthermore, intense cavitational forces may damage delicate tissues like leafy greens, leading to cellular leakage, loss of turgor, and increased nutrient availability for spoilage organisms, potentially accelerating microbial regrowth during storage. Industrial adoption is further hindered by the need for custom-designed equipment and high investment and operating costs associated with generators, transducers, and control systems. These factors present economic and engineering barriers to large-scale integration [14]. Realizing ultrasound's full commercial potential will require advances in equipment standardization, cost reduction, and optimization of treatment protocols tailored to specific food matrices. Collaborative efforts among researchers, equipment manufacturers, and industry stakeholders are essential to overcome these challenges and ensure safe, effective, and scalable implementation of ultrasound in fresh-cut produce preservation.
For pulsed electromagnetic (PEM), to the best of our knowledge, little information is available on the research and application of pulsed electromagnetic in the processing of fresh-cut produce. Furthermore, large-scale implementation of PEM faces economic and technical barriers. Equip-ment costs are high due to the need for specialized power supplies, control systems, and electrodes. Moreover, treatment effectiveness is influenced by food conductivity, chamber geometry, and pulse characteristics, necessitating precise control and cus-tomization. These complexities can hinder adoption, especially in small- and medi-um-scale operations. Therefore, more work is needed to optimize PEM parameters including field strength, pulse number, and waveform across diverse food types.
Based on the above reasons, we did not focus on summarizing the ultrasound and pulsed electromagnetic in this paper. Besides, if these two technologies are added, the length of the article may be too long. However, if necessary, we would be willing to do any revision you demand.
